# Effects of Dietary L-malic Acid Supplementation on Meat Quality, Antioxidant Capacity and Muscle Fiber Characteristics of Finishing Pigs

**DOI:** 10.3390/foods11213335

**Published:** 2022-10-24

**Authors:** Enfa Yan, Yubo Wang, Linjuan He, Jianxin Guo, Xin Zhang, Jingdong Yin

**Affiliations:** State Key Laboratory of Animal Nutrition, College of Animal Science and Technology, China Agricultural University, Beijing 100193, China

**Keywords:** L-malic acid, meat color, carcass traits, antioxidant capacity, muscle fiber type

## Abstract

L-malic acid is a vital intermediate in the citric acid cycle and has been reported to improve the antioxidant capacity and aerobic oxidation of weaned piglets; however, its application in finishing pigs is limited at present. This study explored the effects of dietary L-malic acid supplementation on the carcass traits and meat quality of finishing pigs. In a 45-day experiment, 192 Duroc × Landrace × Yorkshire pigs (75.01 ± 0.51 kg) were divided into four treatments, i.e., a basal diet supplemented with 0, 0.5%, 1%, and 2% L-malic acid complex. The results showed that L-malic acid supplementation had no effects on the growth performance of finishing pigs. Importantly, L-malic acid significantly increased redness (a^*^) value at 24-h postmortem (quadratic, *p* < 0.05) and tended to increase the proportion of oxymyoglobin (OMb) (quadratic, *p* = 0.10), as well as the total antioxidant capacity (T-AOC) activity (quadratic, *p* = 0.08) in the *longissimus dorsi* (LD) muscle. Further, dietary supplementation of 1% L-malic acid markedly increased the protein expression level of slow skeletal myosin heavy chain (MyHC) in the LD muscle (*p* < 0.05). Moreover, 0.5% and 2% L-malic acid supplementation significantly increased carcass length and loin eye area (*p* < 0.05). In conclusion, dietary L-malic acid could effectively improve the meat color and carcass traits of finishing pigs.

## 1. Introduction

Pork is one of the most widely consumed meats in the world, with 109.84 million tons consumed in 2020 [1]. Strikingly, excessive demand for pig growth performance and a lean meat ratio is leading to a decline of pork quality [2]. Intramuscular fat (IMF), drip loss, pH value and meat color are important indicators of the quality of pork. The content of IMF is related to meat edible traits such as juiciness, tenderness and flavor [3]. Drip loss and pH value could affect pork quality by influencing water holding capacity [4,5]. Moreover, as a critical parameter of meat quality, meat color is the primary attribute of sensory perception and directly affects consumers’ desire to purchase [6].

Muscle antioxidant capacity, myoglobin content and muscle fiber type composition are all directly related to meat color [7,8]. Antioxidants can maintain the oxidative stability of meat [9]. Improving muscle antioxidant capacity could reduce drip loss and increase the pH value of pork [10]. From a meat color perspective, altering the antioxidant balance regulates the redox state and perceived color of myoglobin, and intrinsic factors, such as the composition of muscle fiber types, greatly affect the antioxidant balance [11]. According to the morphology, function, and physiological and biochemical characteristics, muscle fibers can be classified as two types: type I (red muscle) and II (white muscle). Type I muscle fibers are the main components of oxidative muscle fibers and express slow myosin heavy chains (MyHCs), which are thinner and richer in myoglobin compared with type II fibers [12,13,14]. Muscle fiber characteristics are reported to affect many pork traits including tenderness, pH value, water holding capacity, and meat color [15,16]. Increasing the amount of slow oxidative muscle fibers can turn the flesh red [11], and thus, contributes to achieving better pork quality [17,18].

L-malic acid is an important intermediate in the citric acid cycle and plays a crucial role in energy metabolism [19]. As a natural plant extract, L-malic acid has antibacterial properties, participates in lipid metabolism and regulates myoglobin redox [20,21,22]. Sodium malate washing solutions were shown to effectively reduce the *Salmonella* content in chicken breast and prolong product shelf life [21]. Another study demonstrated a negative association between muscle malic acid and obesity in rabbits [23]. Our previous study revealed that piglets that were fed L-malic acid during the weaning period had an increased number of oxidative muscle fibers, subsequently improving the water holding capacity of pork without any effects on carcass traits [24]. However, to date, our understanding of the effects of dietary L-malic acid supplementation in finishing pigs on meat quality is still limited. Consequently, the purpose of this study was to investigate the effects and the underlying mechanisms of dietary L-malic acid supplementation on the meat quality and carcass traits of finishing pigs.

## 2. Materials and Methods

### 2.1. Experimental Design and Sample Collection

In our study, 192 crossbred pigs (Duroc × Landrace × Yorkshire, boars and sows ratio was 1:1) with an initial body weight (BW) of 75.01 ± 0.51 kg were housed in pens (1.8 × 2.1 m^2^) and allocated into one of four dietary treatments based on the initial BW, with six replicates (pens) per treatment and eight pigs per pen. The four dietary treatments were basal diet (control) and a basal diet with 0.5%, 1% and 2% L-malic acid complex. The basal diet was formulated according to the National Research Council (NRC, 2012) nutrient requirements for pigs of 75–100 kg BW without any antibiotic additives. All pigs were allowed ad libitum water and feed during the whole trial period (45 d). In the present study, experiments were carried out in the FengNing Swine Research Unit of China Agricultural University (Chengdejiuyun Agricultural and Livestock Co., Ltd., Chengde, China). L-malic acid complex was provided by Anhui Sealong Biotechnology Co., Ltd. (Bengbu, China), which was composed of carrier (80%, mainly zeolite powder) and L-malic acid (20%). The initial BW, final BW and total feed intake of pigs were recorded. Average daily gain (ADG), feed:gain (F:G)and average daily feed intake (ADFI) were calculated. Table 1 shows the ingredient composition and nutrient level of the basal diet.

Pigs with the average final BW (about 110 kg) after treatment (*n* = 8) were selected to fast for 12 h at the end of the trial. We collected blood samples from the precaval vein to obtain plasma, which was stored at −20 °C. Pigs were transported to a modern slaughterhouse (about 1 h), electrically stunned, exsanguinated and eviscerated, in line with the standard commercial procedure, after at least 8 h rest. The carcass was split from the center of the spine. From the right half of each carcass, about 5 g of the *longissimus dorsi* (LD) muscle between the 10th and 12th ribs was sampled and stored at −80 °C.

### 2.2. Carcass Traits

After slaughtering, hot carcass weight was recorded immediately, and dressing percentage was calculated by dividing the hot carcass weight by the final BW. The carcass length and back fat depth at the 6th to 7th rib, last lumbar vertebra, last rib, thickest shoulder and10th rib of the left sides of the carcasses were measured. Loin eye height and width at the 10th rib were measured to calculate loin eye area (Loin eye area (cm^2^) = loin eye height (cm) × width (cm) × 0.7). The fat-free lean index was also calculated by fat-free lean index = 50.767 + [0.035 × hot carcass weight (Ib)] − [8.979 × the last rib fat thickness (in.)] (NRC, 1998).

### 2.3. Meat Quality

Marbling score and the subjective color of the LD muscle were evaluated on the cut surface at the intercostal space between the 10th and 12th ribs according to the NPPC (1999) guidelines. Briefly, the meat color score was based on six color standards (1.0, 2.0, 3.0, 4.0, 5.0 and 6.0), where 1.0 is very pale and 6.0 is dark, purplish red. The marbling score was rated from 1.0 to 10.0, corresponding to the intramuscular fat content. Objective color, including a^*^ (redness), b^*^ (yellowness) and L^*^ (lightness) values, was measured at 45 min and 24 h postmortem using a Colorimeter (CR410, Minolta, Japan). The colorimeter was calibrated against a white tile, according to the manufacturer’s instruction. The pH value was measured at 45 min and 24 h postmortem using a pH meter (Testo 205, Germany). The measurement of drip loss was based on a method described in a previous study [25]; the standard formula for drip loss is as follows: drip loss (%) = [(initial weight − final weight)/initial weight] × 100. To measure cooking loss, a total of 32 muscle samples (about 100 g of per sample) were heated to an internal temperature of 70 °C for 30 min in a thermostatic water bath. The initial and final weight was recorded, and cooking loss was determined by calculating weight change percentage. Then, the dried muscle sample was ground into powder and analyzed for intramuscular (IMF) fat content, as previously described [24].

### 2.4. Texture Characteristics

The LD muscle samples were boiled for 30 min, achieving a central temperature of 70 °C. They were then cooled to room temperature, and samples were trimmed to uniform cubes of about 1 cm^3^. Texture indexes including shear force, hardness, adhesiveness, cohesiveness, springiness, gumminess and chewiness were measured using a Texture Analyzer (TMS-Touch, Food Technology Corp., Sterling, VA, USA), as described previously [24].

### 2.5. Redox Status of Myoglobin in Skeletal Muscle

Samples of about 5 g LD muscle were homogenized with 25 mL of sodium phosphate buffer at a concentration of 0.04 mol/L and pH of 6.8. Homogenization was performed at 10,000 rpm for 20 s at 20 °C, and samples were kept at 4 °C for 1 h, followed by centrifugation at 3500 rpm at 4 °C for 30 min. The supernatant was filtered using filter paper to remove the fat layer [26]. Absorbance at 525 nm, 545 nm, 565 nm and 572 nm was measured by adding the same buffer solution. Relative contents of total myoglobin (TMb), oxymyoglobin (OMb), deoxymyoglobin (DMb)and methemoglobin (MMb) were calculated as follows:TMb mg/g = −0.166A_572_ + 0.086A_565_ + 0.088A_545_ + 0.099A_525_
DMb% = (0.369R1 + 1.140R2 − 0.941R3 + 0.015) × 100
OMb% = (0.882Rl − 1.267R2 + 0.809R3 − 0.361) × 100
MMb% = (−2.541Rl + 0.777R2 + 0.800R3 + 1.098) × 100
where R1, R2 and R3 are the absorbance ratios A572/A525, A565/A525 and A545/A525, respectively [26].

### 2.6. Plasma and Skeletal Muscle Biochemical Parameters

The concentration of lactate and the activities of succinate dehydrogenase (SDH), lactate dehydrogenase (LDH) and malate dehydrogenase (MDH) in the plasma and LD muscle were determined using commercial kits (Nanjing Jiancheng Bioengineering Institute, Nanjing, China). The activities of total superoxide dismutase (T-SOD) and total antioxidant capacity (T-AOC) and the content of malondialdehyde (MDA) in the LD muscle were also determined using commercial kits (Nanjing Jiancheng Bioengineering Institute, Nanjing, China), following the manufacturer’s instructions.

### 2.7. Western Blot Analysis

The total proteins of 0 and 1% L-malic acid complex supplementation groups were extracted using RIPA lysis buffer (Huaxingbio, Beijing, China) and quantified using a BCA protein assay kit (Thermofisher, Waltham, MA, USA). Equal amounts of protein samples (50 ug) were separated by 10% or 8% SDS-PAGE and transferred to Immun-Blot™ polyvinylidene fluoride (PVDF) membranes (Merck Millipore, Darmstadt, German) using BIO-RAD mini protein tetra system. After blocking with a 5% BSA (bovine serum albumin, Huaxingbio, Beijing, China) solution, membranes were incubated overnight with primary antibodies at 4 °C. The membranes were washed in TBST twice and processed with secondary antibody for 60 min at room temperature under dark conditions, before washing in TBST twice and PBS once. Following detection with Odyssey Clx (LI-COR Biotechnology, Lincoln, NE, USA), blots were quantified using ImageJ software (National Institutes of Health, Bethesda, MD, USA). All antibodies used are listed as follows: slow skeletal MyHC (Abcam, Cambridge, MA, USA, ab11083), fast skeletal MyHC (Abcam, ab91506), GAPDH (Cell Signaling Technology, Danvers, MA, USA, #2118), and Anti-rabbit IgG (H + L) (DyLight™ 800 4X PEG Conjugate, Cell Signaling Technology, #5151).

### 2.8. Statistical Analysis

Data were presented as means ± SEM, and analyzed by linear and quadratic regression analysis, unpaired two-tailed Student’s *t*-test or the one-way ANOVA procedures of SAS (v.9.2, SAS Institute, Cary, NC, USA). For growth performance data, each pen was treated as an experimental unit. A value of *p* < 0.05 was considered significant and 0.05 ≤ *p* ≤ 0.10 was considered to have a trend.

## 3. Results

### 3.1. Growth Performance of Finishing Pigs

As shown in Table 2, the addition of L-malic acid in the diet had no effects on the ADG, ADFI or F:G of finishing pigs (*p* > 0.10).

### 3.2. Carcass Traits

As illustrated in Table 3, the addition of L-malic acid to the diet did not alter carcass weight, back fat depth dressing percentage and fat-free lean index. However, carcass length and loin eye area were increased by 0.5% or 2% L-malic acid supplementation relative to the control (*p* < 0.05).

### 3.3. Meat Quality and Texture Profile

L-malic acid significantly increased the a^*^_24 h_ value (quadratic, *p* < 0.05); in particular, dietary supplementation with 1% L-malic acid markedly increased a^*^_24 h_ compared to the control (*p* < 0.05) (Table 4). Other meat quality parameters, including cooking loss, drip loss, IMF content, marbling score and pH values, showed no differences among dietary treatments.

As illustrated in Table 5, dietary supplementation with L-malic acid did not influence the texture characteristics of pork in terms of adhesiveness, chewiness, cohesiveness, gumminess, hardness, shear force or springiness.

### 3.4. Redox Status of Myoglobin

Myoglobin is the sarcoplasmic heme protein which is primarily responsible for meat color [27]. As demonstrated in Table 6, dietary L-malic acid supplementation tended to increase the OMb proportion in the LD muscle (quadratic, *p* = 0.10) but did not affect the TMb content or proportion of DMb and MMb. Importantly, consistent with the a^*^_24 h_ value, dietary 1% L-malic acid supplementation also showed the highest proportion of OMb in the LD muscle.

### 3.5. Lactate Concentration and Metabolic Enzyme Activities

The effects of dietary supplementation with L-malic acid on lactate concentration and metabolic enzyme activities of finishing pigs are summarized in Table 7. L-malic acid tended to elevate the lactate contents in plasma (quadratic, *p* = 0.08), and 1% L-malic acid supplementation exhibited the lowest lactate concentration. The MDH activity in the LD muscle tended to increase in response to 0.5% L-malic acid supplementation (*p* = 0.06), while the LDH and SDH activities were not altered by L-malic acid in the plasma and LD muscle.

### 3.6. Antioxidant Capacity

As shown in Table 8, the T-AOC activity tended to be enhanced by L-malic acid supplementation (quadratic, *p* = 0.08), but T-SOD activity and MDA content were not affected in the LD muscle.

### 3.7. Muscle Fiber Characteristics

As illustrated in Figure 1, considering the best meat quality in the 1% L-malic acid group, muscle fiber type-related proteins of the LD muscle were further detected. The results showed that dietary supplementation of 1% L-malic acid complex significantly increased the protein level of slow MyHC in the LD muscle (*p* < 0.05), while the fast MyHC level was not changed.

## 4. Discussion

With an increasing population and pork consumption around the world, the production of superior quality pork is an inevitable trend [28]. Color is the most important quality attribute of fresh meat influencing purchase decisions [29]. In the present study, dietary L-malic acid improved meat color by increasing the a^*^_24 h_ value of finishing pigs, which was consistent with a previous study which showed that L-malic acid could effectively maintain the redness of bovine *longissimus lumborum* and *psoas major* muscle homogenates [22]. Notably, L-malic acid also tended to increase the ratio of OMb in the LD muscle. Myoglobin is the major pigment affecting the color of fresh meat [11]; it may be present in any of the four redox states, i.e., DMb, OMb, carboxymyoglobin and MMb [29]. Saturating Mb with oxygen provides meat with an attractive cherry-red color by forming OMb [30]. Myoglobin form is the most important factor in the change of a^*^ value [31]. A previous study showed that PUFA-rich plant oils increased OMb content and a^*^ value, similar to our results [32]. Therefore, improved meat color was further evidenced by changes in the chemical characteristics of myoglobin.

Besides the ratio of different myoglobin forms, antioxidant capacity and muscle fiber type composition of skeletal muscle are also crucial factors affecting meat color [33]. Some natural antioxidants, such as grape seed proanthocyanidin extract and lycopene, have been reported to improve the meat color of pork by enhancing antioxidant capacity and increasing slow-twitch fiber percentage [8,34]. Previous studies showed that antioxidants can also reduce the accumulation of MMb and increase the OMb content, further improving meat color [32,35,36]. Moreover, our previous study showed that dietary supplementation of L-malic acid improved the antioxidant capacity of weaned piglets [24]. In this study, dietary L-malic acid supplementation also had a tendency to enhance the T-AOC activity in the LD muscle of finishing pigs. Therefore, we speculate that L-malic acid may increase LD muscle OMb content partly through enhancing antioxidant capacity; however, the effects of L-malic acid supplementation on antioxidant capacity remain to be fully elucidated.

Muscle fiber type is an important determinant of meat color [11]. During muscle development in pigs, an increase in the amount of type I muscle fibers contributes to the redder color of pork [37]. Similarly, the increased expression of *MyHC I* in pork results in increased redness value [38]. Compared with glycolytic fiber types, the main oxidized fiber types have more mitochondria and can promote oxidative metabolism and meat discoloration [39]. Herein, dietary 1% L-malic acid complex supplementation increased the slow MyHC protein level of finishing pigs, indicating the implication of muscle fiber transformation in L-malic acid-improved pork quality. Additionally, an increase in slow MyHC expression level may reduce glycolysis in skeletal muscle, which may result in the lowest lactate concentration in the plasma of the 1% L-malic acid group. Mitochondria is the main site of oxidative phosphorylation. Differences in the content and function of the mitochondria affect the energy metabolism of muscle fibers, as well as the metabolic function of different muscle fibers and, finally, meat color [40]. It still remains to be determined whether L-malic acid improves meat color through enhancing mitochondrial synthesis and function.

Dietary supplementation of 0.5% and 2% L-malic acid significantly increased carcass length and loin eye area in this study. Carcass traits are becoming more important in pork production [27,41]. Carcass length and loin eye area are strongly correlated with economic benefits and carcass value [42]. Dietary supplementation with antioxidants, such as β-glucan, garcinol and resveratrol, has been shown to influence the carcass length or back fat depth of finishing pigs [16,43,44]. It should be noted that thoracolumbar vertebrae number has an important effect on carcass length [45]. Therefore, studies are merited to determine the role of L-malic acid on spine-bone development. Furthermore, multiple studies have attributed the increase in loin eye area to an enhanced protein synthesis process [46,47,48] In this regard, the effects of L-malic acid on protein turnover merits further investigation.

## 5. Conclusions

The present study proved that dietary supplementation with L-malic acid during the finishing period effectively improved the meat color of pigs by elevating slow MyHC expression in the skeletal muscle and enhancing the antioxidant capacity to some extent. Notably, 1% L-malic acid complex is suggested in the diets of finishing pigs to improve meat quality. Meanwhile, dietary supplementation of 0.5% and 2% L-malic acid complex increased the carcass length and loin eye area, thereby improving the carcass characteristics. Our study sheds new light on methods to produce high quality pork and improve its economic value.

## Figures and Tables

**Figure 1 foods-11-03335-f001:**
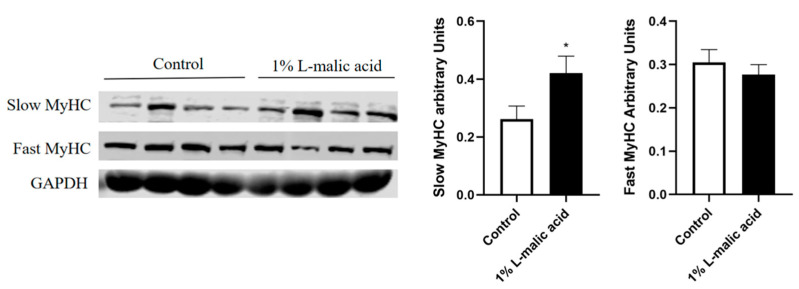
Effects of dietary supplementation with L-malic acid on muscle fiber type composition of the LD muscle (*n* = 8). MyHC, myosin heavy chain. The statistical significance of difference between two means was calculated using *t*-test. * *p* < 0.05.

**Table 1 foods-11-03335-t001:** Ingredients and nutrient levels of diets (%, as-fed basis).

Items	
Ingredients,%	
Corn	81.00
Soybean meal	11.50
Wheat bran	3.20
Soybean oil	1.20
L-Lysine·HCl	0.39
DL-Methionine	0.03
L-Threonine	0.11
L-Tryptophan	0.03
L-Valine	0.02
Limestone	0.70
Calcium hydrogen phosphate	0.90
Salt	0.34
50% Choline chloride	0.08
Premix ^1^	0.50
Total	100.00
Analyzed nutrient levels,%	
Crude protein	12.43
Lysine	0.97
Methionine + cysteine	0.40
Threonine	0.58
Tryptophan	0.14
Isoleucine	0.51
Leucine	1.35
Valine	0.61
Calculated nutrient levels	
DE, Mcal/kg	3429
ME, Mcal/kg	3342
Standardized ileal digestible amino acids,%	
Lysine	0.75
Methionine + cysteine	0.42
Threonine	0.46
Tryptophan	0.13
Isoleucine	0.40
Leucine	1.03
Valine	0.48

Note: ^1^ The premix provided the following per kg of diets: vitamin A, 6000 IU; vitamin D3, 2400 IU; vitamin E, 20 IU; vitamin K3, 2 mg; vitamin B1, 0.96 mg; vitamin B2, 4 mg; vitamin B6, 2 mg; vitamin B12, 0.012 mg; biotin, 0.04 mg; folic acid 0.40 mg; pantothenic acid 11.2 mg; nicotinic acid 22 mg; Cu, 120 mg; Fe, 76 mg; Mn, 12 mg; Zn, 76 mg; I, 0.24 mg; Se, 0.40 mg.

**Table 2 foods-11-03335-t002:** Effects of dietary supplementation with L-malic acid on the growth performance of finishing pigs (*n* = 6).

Items	L-malic Acid Complex Levels (%)				SEM		*p* Value	
	0	0.5	1	2		ANOVA	Linear	Quadratic
Initial BW, kg	75.18	75.29	75.22	75.12	2.98	1.00	0.98	0.98
Final BW, kg	113.43	115.28	113.17	112.66	3.20	0.94	0.74	0.82
ADFI, kg/d	2.90	2.84	2.86	2.76	0.07	0.54	0.18	0.93
ADG, kg/d	0.85	0.89	0.84	0.83	0.01	0.18	0.15	0.32
F: G	3.42	3.20	3.39	3.32	0.09	0.32	0.76	0.57

Note: ADFI, average daily feed intake; ADG, average daily gain; BW, body weight; F:G, feed:gain.

**Table 3 foods-11-03335-t003:** Effects of dietary supplementation with L-malic acid on the carcass traits of finishing pigs (*n* = 8).

Items	L-malic Acid Complex Levels (%)				SEM	*p* Value		
	0	0.5	1	2		ANOVA	Linear	Quadratic
Carcass weight, kg	84.15	83.96	83.73	82.43	1.42	0.83	0.37	0.80
Carcass length, cm	79.06 ^b^	82.91 ^a^	81.46 ^ab^	82.49 ^a^	0.98	0.04	0.07	0.16
Dressing percentage,%	71.13	71.50	71.75	71.50	0.40	0.73	0.54	0.35
Back fat depth, mm								
Shoulder fat thickness	34.58	38.20	37.84	37.31	1.81	0.49	0.43	0.24
The last rib fat thickness	22.40	21.62	21.03	22.56	0.97	0.66	0.83	0.23
Lumbosacral fat thickness	13.98	14.17	14.77	13.08	0.91	0.63	0.47	0.31
The 6th to 7th rib fat thickness	21.03	25.68	24.65	23.83	1.77	0.31	0.48	0.14
The 10th rib fat thickness	18.88	19.69	19.42	17.54	1.25	0.63	0.35	0.37
Average back fat depth	23.24	23.76	24.28	24.60	0.89	0.71	0.27	0.74
Loin eye area, cm^2^	36.87 ^b^	44.63 ^a^	40.66 ^ab^	43.55 ^a^	1.73	0.02	0.06	0.20
Fat-free lean index	49.34	49.41	49.88	49.42	0.29	0.54	0.76	0.25

Note: values with different superscripts indicate significant differences (*p* < 0.05).

**Table 4 foods-11-03335-t004:** Effects of dietary supplementation with L-malic acid on the meat quality of finishing pigs (*n* = 8).

Items	L-malic Acid Complex Levels (%)				SEM	*p* Value		
	0	0.5	1	2		ANOVA	Linear	Quadratic
pH_45 min_	6.35	6.34	6.28	6.22	0.06	0.45	0.16	0.94
pH_24 h_	5.47	5.53	5.54	5.53	0.03	0.33	0.27	0.15
Drip loss,%	1.51	1.36	1.10	1.50	0.20	0.46	1.00	0.14
Cooking loss,%	29.36	30.12	28.65	28.57	1.68	0.91	0.61	0.95
Flesh color score	2.23	2.36	2.41	2.19	0.17	0.76	0.79	0.31
L^*^_45 min_	40.76	40.53	39.90	40.99	0.61	0.62	0.80	0.24
a^*^_45 min_	15.81	16.10	16.11	15.96	0.26	0.82	0.81	0.38
b^*^_45 min_	1.95	2.06	2.00	2.04	0.16	0.96	0.77	0.84
L^*^_24 h_	47.74	46.86	45.76	46.61	0.71	0.28	0.26	0.13
a^*^_24 h_	16.04 ^b^	16.43 ^b^	17.52 ^a^	16.65 ^ab^	0.35	0.04	0.18	0.03
b^*^_24 h_	6.01	5.45	6.02	5.17	0.49	0.53	0.32	0.73
Marbling score	2.15	2.11	2.08	2.31	0.13	0.61	0.35	0.35
Intramuscular fat,%	3.60	4.01	3.70	3.83	0.53	0.82	0.54	0.76

Note: values with different superscripts means significant difference (*p* < 0.05).

**Table 5 foods-11-03335-t005:** Effects of dietary supplementation with L-malic acid on the LD muscle texture characteristics of finishing pigs (*n* = 8).

Items	L-malic Acid Complex Levels (%)				SEM	*p* Value		
	0	0.5	1	2		ANOVA	Linear	Quadratic
Shear force, N	65.21	71.65	71.20	66.10	4.61	0.67	0.92	0.24
Hardness, N	29.98	31.37	31.41	30.31	2.15	0.95	0.99	0.57
Adhesiveness, mJ	0.08	0.09	0.09	0.08	0.01	0.87	1.00	0.44
Cohesiveness, Ratio	0.53	0.53	0.53	0.53	0.008	0.89	0.94	0.45
Springiness, mm	2.76	2.90	2.74	2.77	0.07	0.39	0.66	0.65
Gumminess, N	15.01	16.82	16.73	17.02	1.12	0.57	0.29	0.44
Chewiness, mJ	44.22	45.60	46.85	48.31	4.35	0.92	0.50	0.91

**Table 6 foods-11-03335-t006:** Effects of dietary supplementation with L-malic acid on the LD muscle myoglobin of finishing pigs (*n* = 8).

Items	L-malic Acid Complex Levels (%)				SEM	*p* Value		
	0	0.5	1	2		ANOVA	Linear	Quadratic
DMb, %	44.00	42.79	41.86	42.20	1.10	0.55	0.27	0.36
MMb, %	19.01	19.41	18.05	20.00	0.98	0.56	0.12	0.84
OMb, %	18.41	19.43	21.75	19.35	1.25	0.30	0.57	0.10
TMb, mg/g	0.013	0.014	0.014	0.015	0.00	0.64	0.29	0.70

Note: DMb, deoxymyoglobin; MMb, mthemoglobin; OMb, oxymyoglobin; TMb, total myoglobin.

**Table 7 foods-11-03335-t007:** Effects of dietary supplementation with L-malic acid on lactate content and metabolic enzyme activities of finishing pigs (*n* = 8).

Items	L-malic Acid Complex Levels (%)				SEM	*p* Value		
	0	0.5	1	2		ANOVA	Linear	Quadratic
Plasma								
Lactate, mmol/L	4.48	4.36	4.10	4.95	0.24	0.16	0.18	0.08
LDH, U/L	626.60	645.95	619.49	640.38	31.00	0.93	0.89	0.90
MDH, U/mL	1.58	1.51	1.41	1.46	0.06	0.22	0.15	0.16
SDH, U/mL	17.53	22.36	20.26	18.69	2.17	0.44	0.96	0.20
LD muscle								
Lactate, mmol/g	9.02	10.22	8.36	9.54	0.72	0.33	0.96	0.73
LDH, U/gprot	552.02	528.15	525.44	523.85	11.02	0.25	0.12	0.23
MDH, U/mgprot	0.55	0.72	0.58	0.53	0.06	0.06	0.25	0.13
SDH, U/mgprot	7.48	7.45	6.82	7.41	0.45	0.69	0.83	0.37

Note: LD, *longissimus dorsi*; LDH, lactate dehydrogenase; MDH, malate dehydrogenase; SDH, succinate dehydrogenase.

**Table 8 foods-11-03335-t008:** Effects of dietary supplementation with L-malic acid on the antioxidant capacity of the LD muscle in finishing pigs (*n* = 8).

Items	L-malic Acid Complex Levels (%)				SEM	*p* Value		
	0	0.5	1	2		ANOVA	Linear	Quadratic
MDA, nmol/mgprot	0.10	0.11	0.12	0.08	0.01	0.25	0.20	0.13
T-AOC, U/mgprot	0.10	0.14	0.13	0.11	0.02	0.32	0.85	0.08
T-SOD, U/mgprot	90.47	101.92	95.05	77.63	7.76	0.19	0.13	0.14

Note: MDA, malondialdehyde; T-AOC, total antioxidant capacity; T-SOD, total superoxide dismutase.

## Data Availability

Data is contained within the article and available upon request from the corresponding author.

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
