# Peer review of "Effects of Dietary L-malic Acid Supplementation on Meat Quality, Antioxidant Capacity and Muscle Fiber Characteristics of Finishing Pigs"

_foods, 2022, doi:10.3390/foods11213335_

Round 1
Reviewer 1 Report
Thank you for this relevant paper. Some comments:
line 44: MyHC pops up as an acronym here. please explain acronyms first, before using them as such in the text. What you could do here is: "myosine heavy chain (MyH)". From then on, you can use the acronym,
line 50-51: "L-malic acid is of antibacterial properties" should read "L-malic acid has antibacterial properties".
line 91: "LD muscle": again an acronym. Better to write " Latissimus dorsi (LD) muscle"
line 162: what is the reason that three separate statistical analyses are used on the same material? One appropriate method would have sufficed? Particularly since conclusions are drawn about significance based on a significant result of only one (probably the most appropriate method in this case) of the statistical methods (table 3).
Reviewer 2 Report
L 27-28 In the last 5 yrs poultry and not pig is the most consumed meat in the world https://ourworldindata.org/grapher/global-meat-production-by-livestock-type?time=1961..2020
Therefore, please change to read: Pork is of the most consumed meat …..
and change accordingly the reference.
L103-104 please explain in detail how did you measure the subjective color and marbling in order to provide all the required information to someone to repeat the experiment.
L106-107 As far as I am aware, calibration should be applied against the black tile as well. Please check the instructions again.
L114 What is the: free-dried muscle power…??
Tables 6, 7 and 8 and figure 1. Tables should stand alone. Please provide explanations for all abbreviations in footnote.
L 255-256 and Conclusions section. You speculate that the antioxidant capacity has been improved but this statement is not justified by your results. No mean differences have been obtained apart from a tendency for a quadratic effect. Please revise accordingly.
L272-277 The increase in carcass length is very interesting and I must say unexpected. Therefore, I consider that you should further discuss it by taking into consideration that carcass length is affected more by spine-bone development rather than muscle development. I consider that your statement that muscle growth affected carcass length is not that rational. In addition, there is a vast number of published articles that haven’t found any effect of antioxidants supplementation on carcass traits. Please discuss it in the discussion section.
